# Insights into the Mechanism and Catalysis of Peptide Thioester Synthesis by Alkylselenols Provide a New Tool for Chemical Protein Synthesis

**DOI:** 10.3390/molecules26051386

**Published:** 2021-03-04

**Authors:** Florent Kerdraon, Gemma Bogard, Benoît Snella, Hervé Drobecq, Muriel Pichavant, Vangelis Agouridas, Oleg Melnyk

**Affiliations:** 1U1019-UMR 9017—CIIL—Center for Infection and Immunity of Lille, Institut Pasteur de Lille, University of Lille, CNRS, Inserm, CHU Lille, F-59000 Lille, France; florent.kerdraon@ibl.cnrs.fr (F.K.); gemma.bogard@inserm.fr (G.B.); benoit.snella@ibl.cnrs.fr (B.S.); herve.drobecq@ibl.cnrs.fr (H.D.); muriel.pichavant@pasteur-lille.fr (M.P.); 2Centrale Lille, F-59000 Lille, France

**Keywords:** peptide thioester, alkylselenol, catalysis, chemical protein synthesis

## Abstract

While thiol-based catalysts are widely employed for chemical protein synthesis relying on peptide thioester chemistry, this is less true for selenol-based catalysts whose development is in its infancy. In this study, we compared different selenols derived from the selenocysteamine scaffold for their capacity to promote thiol–thioester exchanges in water at mildly acidic pH and the production of peptide thioesters from *bis*(2-sulfanylethyl)amido (SEA) peptides. The usefulness of a selected selenol compound is illustrated by the total synthesis of a biologically active human chemotactic protein, which plays an important role in innate and adaptive immunity.

## 1. Introduction

Chemical protein synthesis is now established as an alternative mode of production of tailored proteins for use in diverse fields of research, primarily biological and medicinal research [1]. Thioester chemistry is central to the field of chemical protein synthesis by the frequent application of the native chemical ligation (NCL [2,3,4,5,6], Figure 1A) and desulfurization methods [7,8,9,10] to the chemoselective formation of peptide junctions to cysteine or alanine in water using unprotected peptide segments as reactants [11].

Besides the assembly of the protein itself and the acceleration of peptide bond formation [12,13,14,15,16], one limitation of chemical protein synthesis using NCL is the production of the peptide thioester segments [17]. Therefore, facilitating the access to the peptide thioesters using Fmoc-SPPS has been intensively pursued, especially during the last decade, and the progress made in this area explains in part the rapid increase in the number of protein targets assembled since 2010 [1,11]. One corpus of methods that has contributed to the progress of chemical protein synthesis exploits the capacity of *N*-(2-mercaptoethyl)amides, i.e., *N,S*-acyl shift systems [18,19,20,21,22,23,24,25,26,27,28,29,30,31,32,33,34,35,36], to rearrange into thioesters in aqueous media. Several of these systems have been validated by the synthesis of challenging proteins [37,38,39,40,41,42,43,44].

One popular mode of use of *N,S*-acyl shift systems is the production of alkylthioesters by exchanging the *N*-(2-mercaptoethyl)amine unit by a thiol used in excess, i.e., R’SH in Figure 1B. Such exchange reactions are equilibrated, the equilibrium being displaced toward the thioester form at acidic pH probably due to the masking of the departing *N*-(2-mercaptoethyl)amine by protonation [45]. Detailed mechanistic studies performed on the *bis*(2-sulfanylethyl)amido (SEA) system established that in such conditions the thiol–thioester exchanges are rate limiting. This is because thiol–thioester exchanges proceed preferably through thiolate species [46], while the attacking thiol R’SH in the process shown in Figure 1B is mostly in its neutral form at the acidic pH used for the exchange reaction, typically pH 4 [47]. These mechanistic findings led to the conclusion that the production of peptide thioesters from SEA peptides might be facilitated by catalyzing the thiol/thioester exchange step shown in Figure 1B. Indeed, efficient catalysis could be achieved by using diselenol catalyst **8a** obtained by in situ reduction of diselenide **7a** (R” = alkyl group, Figure 1D), which in practice was found to be superior to other classical thiol or selenol additives for promoting the production of peptide thioesters from SEA peptides at pH 4. The design of catalyst **8a** has been inspired by the observation that the simpler diselenol **8b** (R” = H, Figure 1D) was able to catalyze the SEA/thiol exchange reaction, albeit with byproduct formation due to the presence in **8b** of a nucleophilic secondary amine, hence the introduction of a blocking alkyl group on the nitrogen.

The diverse kinetic data collected so far support that the catalysis of the SEA/thiol exchange reaction by selenols proceeds as shown in Figure 2 [45,47]. According to this mechanism, the SEA peptide **9** undergoes a spontaneous *N,S*-acyl shift to produce the transient SEA thioester **10**. The latter reacts with the selenol catalyst probably through selenoate species to produce selenoester **12**, which then undergoes an exchange reaction with the thiol additive used in excess. The catalytic potency of selenols stems from their low pK_a_ compared to thiols, which ensures that a significant fraction of the selenol is in the form of a nucleophilic selenoate at the working acidic pH for the reaction, and also from the strong acylating properties of selenoesters of type **12** [13,16].

The ease of synthesis, the catalytic potency and the water solubility are certainly key points to consider in the design of catalysts acting on unprotected peptide segments. Diselenol catalysts of type **8a** are efficient and water soluble but their synthesis is complex and low yielding. This latter limitation prompted us to search for novel selenol catalysts whose production would be easier. Evolving catalyst **8a** structure into easier to make selenol catalysts while maintaining similar catalytic potencies is not obvious. This is because the importance of the bis selenol and tertiary amine functionalities in the catalysis by **8a** remains unknown. In addition, the consequences of changing catalyst structure on selenol nucleophilicity and pK_a_ are hard to predict.

In this work, we evaluated the interest of selenol compounds **13** and **14** for catalyzing the SEA/thiol exchange process (Figure 3). These compounds, which share the 2-(selanylethyl)amino motif with original catalyst **8a**, have been studied in the past as analogs of choline [48,49,50]. They have the advantage of being easily produced in multigram quantities from cheap starting materials. Detailed kinetic studies showed the capacity of selenol **13** to act as a substitute of diselenol **8a** for the synthesis of peptide thioesters. Its usefulness is illustrated by the first chemical synthesis of a biologically active 9 kDa granulysin protein, an antibacterial human protein secreted by cytotoxic cells in response to various infectious agents [51,52,53].

## 2. Results and Discussion

### 2.1. Catalyst Synthesis

Catalysts **13** and **14** were produced as their corresponding diselenides, i.e., diselenides **17** and **18** respectively, starting from 2-chloro-*N,N*-dimethylethylamine hydrochloride according to known procedures [50,54]. Briefly, 2-chloro-*N,N*-dimethylethylamine was neutralized with potassium carbonate and then reacted with potassium selenocyanate to produce *N*,*N*-dimethyl-2-selenocyanatoethan-1-amine **16**, which upon treatment with sodium hydroxide afforded diselenide **17** (Scheme 1). Diselenide **17** was easily alkylated with methyl iodide to produce diselenide **18**.

### 2.2. Kinetic Studies

The SEA/thiol exchange reaction used for studying the potency of the different catalysts is shown in Scheme 2. The water soluble *tris*(2-carboxyethyl)phosphine (TCEP) enables an in situ reduction of the cyclic SEA^off^ disulfide **19** and of the diselenide precatalyst **17** or **18**. The thiol additive used in this reaction is 3-mercaptopropionic acid (MPA), which is a classical thiol used for preparing peptide thioesters.

The exchange reactions were monitored by HPLC (Figure 4A). The kinetic data were fitted to extract the apparent second order rate constants (Figure 4B). In this figure and in the following paragraphs, the data are expressed as a function of total selenol concentration. As suggested by the experimental data shown in Figure 4B, the rate constants obtained for **13** and **8a,** which are formed by in situ reduction of diselenides **17** and **7a**, plateau at the same value above 50 mM. However, examination of Figure 4B or of Table 1, which gives the half-reaction times for the different experiments, shows that the potencies of catalysts **13** and **8a** are not the same for concentrations below 50 mM. In such a case, the SEA/thiol exchange reaction proceeds faster with catalyst **8a**. The difference is particularly striking by looking at the data obtained at the lowest catalyst concentration tested (compare in the insert of Figure 4A blue circles and triangles for **13** and **8a** respectively used at 6.25 mM concentration). We can conclude at this point that catalyst **8a** can be substituted by selenol **13** with the condition of being used at a minimal concentration of 50 mM, under which conditions the highest rate for the SEA/thiol exchange reaction can be achieved.

The ability to accelerate the SEA/MPA exchange reaction was also assessed according to the same conditions for selenol **14** obtained from in situ reduction of **18**. Whether at a concentration of 200 mM, 100 mM or 50 mM, selenol **14** was found to be significantly less potent than catalysts **8a** and **13** (Table 1). We noticed the formation of a red solid in the reaction mixture after a few hours at the highest concentration tested (200 mM), probably elemental selenium (see Appendix A). This observation strongly suggests that catalyst **14** decomposes throughout the course of the reaction. We also noticed an increase of the pH of the reaction mixture, a phenomenon that was not observed with the other selenols. Since previous studies established that the SEA/thioester ratio at equilibrium increases with pH [45,47], the pH drift observed in these experiments might explain in part the diminished conversion observed with catalyst **14**. Taken together, the lower conversion and rates observed with catalyst **14** preclude its use as an interesting alternative to **8a** for the production of peptide thioesters from SEA peptides.

Another striking observation made during the HPLC-MS monitoring of the exchange process catalyzed by selenol **14** was the formation of the peptide selenoester derived from selenocholine in substantial amounts (Figure 4C). In contrast, the amount of selenoester formed during the SEA/thiol exchange catalyzed by selenols **8a** or **13** was below HPLC-MS detection limits. The persistence of the selenoester derived from **14** in the reaction mixture suggests that the quaternarization of the basic nitrogen significantly alters the reactivity of the carbonyl group nearby. This observation is reminiscent of the work of Mautner and coworkers, who noticed that the hydrolysis of 2-dimethylaminoethyl selenobenzoate proceeded significantly faster at pH 7 than hydrolysis of benzoylselenocholine. The authors proposed that the formation of an intramolecular hydrogen bond between the protonated secondary amine and the selenoester carbonyl oxygen might promote the hydrolysis of the selenoester carbonyl [49]. Such an intramolecular activation of the selenoester group by the protonated amine as illustrated in Figure 5 might contribute to the catalytic potency of selenols **8a** and **13**.

### 2.3. Total Chemical Synthesis of 9 kDa Granulysin

To illustrate the usefulness of the new catalyst **13**, we undertook the total chemical synthesis of 9 kDa granulysin (9-GN, Figure 6A). 9-GN is a human cytotoxic and proinflammatory protein that belongs to the NK-lysin family of proteins [53,55,56]. The protein is stabilized by two disulfide bonds. 9-GN protein is secreted by specialized cells from the immune system in response to infection by various agents such as bacteria, parasites, fungi and viruses. Designing an efficient and modular synthetic approach toward 9-GN can facilitate the development of novel antibacterial proteins of potential therapeutic interest.

The synthetic strategy that was followed for accessing the linear 9-GN polypeptide (**9-GN-l**) is described in Figure 6B. **9-GN-l** polypeptide was assembled from three peptide segments using NCL, which are highlighted by orange (N-terminal), blue (middle) and green (C-terminal) colors in Figure 6. The preceding residue of all the cysteines present in 9-GN sequence are β-branched amino acid residues, meaning that regardless of the AA-Cys junctions chosen for 9-GN assembly, the formation of difficult peptidic junctions will have to be dealt with. The assembly was done in the C-to-N direction and therefore required the protection of the N-terminal cysteine residue of the middle segment to avoid its cyclization or oligomerization during the first ligation step. The acetoacetyl (AcA) group was chosen for this purpose due to its ease of installation and removal [37,38,39,57].

The preparation of the middle thioester segment **21** in blue in Figure 6B proved to be challenging. First, the classical Fmoc SPPS of the SEA peptide precursor resulted in low yields due to difficult couplings after Arg94. This problem was solved by introducing Ala92 and Thr93 as a pseudoproline dipeptide unit, i.e., Fmoc-Ala-Thr(Ψ(Me,Me)Pro-OH (Figure 7) [58]. The second issue was the sluggish SEA/MPA exchange process enabling the installation of the thioester group, due to the presence of a sterically demanding valine residue at the C-terminus. Catalyst **13** proved to be particularly useful in this case. Interestingly, the protection of the N-terminal cysteine residue using methyl-*S*-(acetoacetyl)-thioglycolate (AcA-MTG) in the presence of 4-mercaptophenylacetic acid (MPAA) and the diselenide-catalyzed SEA/MPA exchange reaction could be performed in one-pot with a pH change in between, thereby saving one intermediate purification step.

The successful synthesis of thioester segment **21** set the stage for the assembly of the linear **9-GN-l** polypeptide (Figure 8). The ligation of middle **21** and right **22** segments was followed by the removal of the AcA group in one pot by adding hydroxylamine hydrochloride to the reaction mixture and adjusting the pH to mildly acidic values. The second ligation step with peptide thioester **24** provided the full length **9-GN-l** polypeptide which was purified by HPLC.

The folding step was performed at 20 °C in a phosphate buffered saline/glycerol 9/1 mixture in the presence of glutathione (1 mM)/glutathione disulfide (0.2 mM) redox system and guanidine hydrochloride as a solubilizing additive. The use of guanidine hydrochloride during this step was mandatory to enable the solubilisation of the linear **9-GN-l** polypeptide. The formation of the native pattern of disulfide bonds as shown in Figure 6A has been demonstrated by identifying the peptides produced upon trypsin digestion under non-reducing conditions by mass spectrometry (see Appendix A). The UPLC-MS analysis of **9-GN** shown in Figure 9 highlights the quality of the protein obtained by the designed synthetic route.

### 2.4. Biological Activity of Synthetic 9-GN

The successful synthesis and folding of 9-GN set the stage for examining its biological activity. One hallmark of 9-GN is its capacity to induce the migration of immune cells to the site of infection. Typically, 9-GN is a chemoattractant for monocytes in the low nanomolar range in a transwell migration assay performed in Boyden micro chambers.

In such an assay, monocytes are placed on the upper layer of a cell permeable membrane inserted in a multi-well plate, which separates the cells from the solution containing the chemoattractant. After incubation (1 h), the cells that have migrated through the membrane are stained and counted. In such an assay, the synthetic 9-GN protein displayed a strong migration index in the 1–100 nM range as expected from literature data (Figure 10), showing the biological functionality of the protein obtained by chemical synthesis [56].

## 3. Materials and Methods

### 3.1. Synthesis of Diselenide Precatalyst 17

To a solution of potassium carbonate (11.5 g, 0.083 mol) in water (30 mL), was added 2-chloroethyl-*N*,*N*-dimethylamine hydrochloride (6.0 g, 0.042 mol). The resulting mixture was stirred at room temperature for 10 min. The resulting crude mixture was extracted 3 times with 30 mL of Et_2_O, dried over MgSO_4_ and concentrated under reduced pressure. 1.0 g of the resulting pale yellow oil (9.3 mmol) was diluted in 90 mL of acetonitrile and potassium selenocyanate (1.47 g, 10.2 mmol) was added portionwise. The resulting solution was stirred at room temperature for 16 h. Et_2_O (100 mL) was added to the reaction mixture which was then filtered through a pad of celite. The filtrate was evaporated in vaccuo and then redissolved in 120 mL of absolute ethanol. Aqueous sodium hydroxide (60 mL, 1 M) was slowly added under stirring and the reaction mixture was stirred 3 h at room temperature. The resulting mixture was concentrated in vacuo to remove the maximum of ethanol and extracted Et_2_O (3 × 30 mL). The organic layers were combined, dried over MgSO_4_ and concentrated under reduced pressure to afford diselenide **17** as a yellow oil (1.24 g, 4.09 mmol, 88%).

^1^H NMR (300 MHz, CDCl_3_) δ (ppm) 3.10–2.96 (m, 4H), 2.68–2.55 (m, 4H), 2.25 (s, 12H) (see Appendix A).

^13^C NMR (75 MHz, CDCl_3_) δ (ppm) 60.32, 45.20, 28.20 (see Appendix A).

MALDI-TOF Matrix 2,5-dihydrobenzoic acid, positive detection mode, calcd. for [M+H]^+^ (exact mass): 305.00, found: 304.92.

### 3.2. Synthesis of 9-GN

Step 1. Synthesis of peptide thioester **21**

To a solution of Gn·HCl (1.1 g) in 0.1 M pH 7.4 phosphate buffer (1.2 mL) was added MPAA (17 mg, 50 mM). SEA^off^ peptide **26** (15 mg, 2.6 μmol, 1.0 equiv) and AcA-MTG (77 µL of a 0.10 M solution in ACN, 7.7 μmol, 3.0 equiv) were then successively dissolved in the MPAA solution (1.3 mL) and the pH of the mixture was adjusted to 7.2 by addition of 6 M NaOH. The mixture was stirred at 37 °C for 8 h. 3.0 additional equivalents of AcA-MTG were added and the mixture was stirred an additional 16 h at 37 °C under nitrogen atmosphere. Completion of the reaction was ascertained by UPLC-MS analysis.

After protection of the N-terminal Cys residue (24 h), the subsequent SEA/MPA exchange reaction was carried out in one pot. TCEP∙HCl (34 mg, 0.10 M), diselenide catalyst **17** (20 mg, 50 mM) and MPA (65 μL, 5% by vol) were added to the previous mixture. The pH was adjusted to 4.0 by addition of 6 M NaOH and the reaction mixture was stirred at 37 °C for 72 h. After completion of the reaction, the mixture was diluted in 10 mL of water containing 0.1% TFA. The crude was purified by HPLC (eluent A = water containing 0.1% TFA, eluent B = acetonitrile in water 4/1 *v*/*v* containing 0.1% TFA, 50 °C, detection at 215 nm, 6 mL min^−1^, 0–37% eluent B in 15 min, then 37–57% eluent B in 60 min, C18XBridge column) to give peptide thioester **21** as a white solid after lyophilization (5.09 mg, 34%).

Peptide **21** was analyzed by UPLC-MS (see Appendix A) and MALDI-TOF mass spectrometry (see Appendix A).

ESI (positive detection mode, see Appendix A): calcd. for [M] (average): 4561.40, found: 4561.79.

MALDI-TOF (positive detection mode, matrix 2,5-dihydrobenzoic acid, see Appendix A): calcd. for [M + H]^+^ (average): 4562.40, found: 4562.87.

Step 2. Synthesis of peptide amide **23**

To a solution of Gn·HCl (0.57 g) in 0.1 M, pH 7.4 phosphate buffer (0.60 mL) was added MPAA (34 mg, 0.20 mmol) and the pH of the mixture was adjusted to 7.2 by addition of 6 M NaOH. Peptide **21** (3.5 mg, 0.61 µmol, 1.0 equiv) and peptide **22** (2.8 mg, 0.67 μmol, 1.1 equiv) were then successively dissolved in the MPAA solution (0.12 mL) and the pH of the mixture was adjusted to 7.2 by addition of 6 M NaOH. The mixture was stirred at 37 °C for 24 h under nitrogen atmosphere. Completion of the reaction was insured by UPLC-MS analysis.

After completion of the NCL ligation (24 h), the AcA protecting group was removed by reaction with NH_2_OH. The reaction mixture was first acidified to pH 3.0 with 6 M HCl and extracted 3 times with 1.0 mL of Et_2_O to remove MPAA. A 0.5 M solution of NH_2_OH·HCl in H_2_O (26 μL, 13 μmol, 20 equiv) was then added and the pH was adjusted to 5.5. After 4 h stirring at 37 °C, pH was adjusted to 3.0 with 0.25 g mL^−1^ aqueous solution of TCEP∙HCl. The crude was further diluted in 4.0 mL of water containing 0.25% TFA and purified by HPLC (eluent A = water containing 0.25% TFA, eluent B = acetonitrile in water 4/1 *v*/*v* containing 0.25% TFA, 50°C, detection at 215 nm, 6 mL min^−1^, 0–40% eluent B in 10 min, then 40–50% eluent B in 45 min, C18XBridge column) to afford peptide **23** as a white solid after lyophilization (3.49 mg, 56%).

Peptide **23** was analyzed by UPLC-MS (see Appendix A) and MALDI-TOF mass spectrometry (see Appendix A).

ESI (positive detection mode, see Appendix A): calcd. for [M] (average): 7899.22, found: 7899.94.

MALDI-TOF (positive detection mode, matrix 2,5-dihydrobenzoic acid, see Appendix A): calcd. for [M+H]^+^ (average): 7900.22, found: 7901.49.

Step 3. Synthesis of **9-GN-l**

To a solution of Gn·HCl (0.57 g) in 0.1 M pH 7.4 phosphate buffer (0.60 mL) was added MPAA (34 mg, 0.20 mmol) and the pH of the mixture was adjusted to 7.7 by addition of 6 M NaOH. Peptide **23** (3.5 mg, 0.36 μmol, 1.0 equiv) and peptide thioester **24** (0.38 mg, 0.36 μmol, 1.0 equiv) were then successively dissolved in the MPAA solution (91 µL) The mixture was stirred at 37 °C for 24 h under nitrogen atmosphere. Completion of the reaction was insured by UPLC-MS analysis.

After completion of the NCL ligation (24 h), TCEP∙HCl (5.2 mg, 18 µmol) was added to the reaction medium to reduce any mixed disulfide formed during ligation. The resulting mixture was diluted in 3.0 mL of water containing 0.1% TFA. The crude was purified by HPLC (eluent A = water containing 0.1% TFA, eluent B=acetonitrile in water 4/1 *v*/*v* containing 0.1% TFA, 50 °C, detection at 215 nm, 6 mL min^−1^, 0–38% eluent B in 10 min, then 38–58% eluent B in 60 min, C18XBridge column) to give **9-GN-l** as a white solid after lyophilization (2.92 mg, 78%).

Peptide **9-GN-l** was analyzed by UPLC-MS (see Appendix A) and MALDI-TOF mass spectrometry (see Appendix A).

ESI (positive detection mode, see Appendix A): calcd. for [M] (average): 8435.79, found: 8435.45.

MALDI-TOF (positive detection mode, matrix 2,5-dihydrobenzoic acid, see Appendix A): calcd. for [M + H]^+^ (average): 8436.79, found: 8436.65.

### 3.3. Folding

The linear polypeptide **9-GN-l** (0.558 mg, 0.054 µmol) was dissolved in PBS containing 10% by vol of glycerol, 2 M of Gn·HCl, 1 mM reduced glutathione, 0.2 mM oxidized glutathione (1 mL/mg final peptide concentration). The reaction mixture was let 1 h at 0 °C, then 8 days at 20 °C.

The protein was dialyzed twice (respectively 2 h and 24 h) at 20 °C against 200 mL PBS containing 10% in vol of glycerol. 0.71 mL of the dialyzed solution were recovered and the concentration of the folded protein was determined using UV quantification at 280 nm (60.7 µM, 0.043 µmol, 80%)

Folded protein **9-GN** was analyzed by UPLC-MS (see Appendix A).

Experimental determination of the disulfide bridge pattern was achieved by trypsin digestion of **9-GN** and identification of the resulting fragments by mass spectrometry using non-reducing conditions. Fragments obtained by digestion with trypsin permitted the direct assignment of Cys69-Cys132 and Cys96-Cys107 disulfide bonds (see Appendix A).

### 3.4. Chimiotaxis Assay

Blood monocytes were purified from six donors by positive selection over a MACS column using anti-CD14-conjugated microbeads (Miltenyi Biotec) as previously described [60].

Monocyte migration (5 × 10^4^ cells/well in 50 μL of RPMI 1640 medium with 0.1% FCS) was performed in a 48-well Boyden microchamber (Neuroprobe) through a standard 5-μm pore filter (Neuroprobe), at 37 °C for 1 h. Migration was evaluated in response to synthetic **9-GN** (1 to 100 nM in RPMI 1640 medium containing 0.1% FCS), CCL2 used as positive control (200 ng/mL, 23.1 nM, R&D Systems), and medium alone used as negative control. The filter was stained using Diff–Quik reagent. Migrated cells were counted on the lower side of the filter in three randomly selected high power fields (magnification × 100). Each assay was performed in triplicate. Results are expressed as the difference between mean numbers of cells per high power fields minus the negative control (medium alone).

## 4. Conclusions

We conducted detailed kinetic studies of the catalysis of the *bis*(2-sulfanylethyl)amido (SEA)/thiol exchange process by selenols derived from selenocysteamine. These data provide important clues regarding the role played by the basic amino group of the selenocysteamine scaffold in catalysis and how to evolve the catalyst to achieve useful catalytic potencies while minimizing synthetic cost. Catalysis by *bis*(*N*,*N*-dimethylaminoethyl)diselenide proved particularly useful and involves presumably an intramolecular general acid catalysis by the proton of the dimethylammonium group. *Bis*(*N*,*N*-dimethylaminoethyl)diselenide can be easily produced in multigram scale from cheap starting materials. Its usefulness is illustrated by the chemical synthesis of a biologically actice 9 kDa granulysin.

## Data Availability

Not applicable.

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
