# Peer review of "Insights into the Mechanism and Catalysis of Peptide Thioester Synthesis by Alkylselenols Provide a New Tool for Chemical Protein Synthesis"

_molecules, 2021, doi:10.3390/molecules26051386_

Round 1

Reviewer 1 Report

The article Insights into the mechanism and catalysis of peptide thioester synthesis by alkyselenols provide a new tool for chemical protein synthesis deals about the synthesis of new amino-selenols used as catalysts for an N,S acyl type of reaction : a SEA (bis(2-sulfanylethyl)amido)/thiol reaction. 

The author of the article seems to be expert in the field of the peptide synthesis and the field around this. They show how they synthesized the catalysts and after that, they give more evidence that efficiency of the  catalyst in link to the formation of intramolecular hydrogen bond between the amino group and the carbonyl group during this exchange.

------------------------------------------------------------------------------------The compounds 13 and 14 were not strictly isolated and tried as catalysts, even with the air instability of theses compouds, some isolation will have been tried. 

------------------------------------------------------------------------------------The part 3.Materials and Methods is not well adapted. 
It must be placed at the begining, before 2.results and discussion or after conclusion, but moreover they described synthesis of only one step of the catalysts synthesis, whereas the other step are in the supplementary information. Please harmonised this (everything in the SupInfo or all the procedure in the article, but not a mixture.

------------------------------------------------------------------------------------

In the Supporting Information, it is the same :

p19 in the H NMR of 17, the integration is done, by considerating that the molecule is symmetric (so there are 10H in the spectra and 20 H in total, because molecule is symmetric). But in the H spectra of 18, the authors put the integration, without considerating an symmetry ( 26 H in the spectra for 26 H in the molecule)

Please harmonize, moreover the spectra need to be corrected. 

------------------------------------------------------------------------------------

For me, JMOD doesnt exist, it is DEPT 135. 

------------------------------------------------------------------------------------

Considering these changes, the article can be published.

Author Response

Dear Referee,

We thank you for helping us improving our manuscript.

Our response to your comments and suggestions can be found below within your report.

Best Regards

Oleg Melnyk & Vangelis Agouridas

The compounds 13 and 14 were not strictly isolated and tried as catalysts, even with the air instability of theses compouds, some isolation will have been tried.

Response:

Isolation of the catalysts in their selenol form is pointless in the present case. Due to their known instability, this would result in significant distorsions in the evaluation of their catalytic potency and additional difficulties for their characterization.

Response:

It can actually be considered as a strength to provide precatalysts in the form of diselenides that can be stored over long time periods and that are activatable on demand.

The part 3.Materials and Methods is not well adapted. 
It must be placed at the begining, before 2.results and discussion or after conclusion, but moreover they described synthesis of only one step of the catalysts synthesis, whereas the other step are in the supplementary information. Please harmonised this (everything in the SupInfo or all the procedure in the article, but not a mixture.

Response:

We took the decision to focus the Materials and Methods section to the most important protocols, that is the synthesis of precatalyst 17 and of 9-GN protein. The Supplementaty Information file is huge and therefore it can hardly fit into the main manuscript. Alternately, removing all experimental details from the main manuscript would oblige the reader to go systematically through the Supplementary Information file, which is not optimal.

Although we understand the comment of this Referee, we would like to keep the Materials and Methods section as is.

------------------------------------------------------------------------------------

In the Supporting Information, it is the same:

p19 in the H NMR of 17, the integration is done, by considerating that the molecule is symmetric (so there are 10H in the spectra and 20 H in total, because molecule is symmetric). But in the H spectra of 18, the authors put the integration, without considerating an symmetry ( 26 H in the spectra for 26 H in the molecule)

Please harmonize, moreover the spectra need to be corrected. 

Response:

This has been corrected considering all the protons in the molecules (see the NMR spectrum of 17 – figure S12 - in the SI).

------------------------------------------------------------------------------------

For me, JMOD doesnt exist, it is DEPT 135. 

Response:

The NMR experiments used in this case are J-MOD (J-modulated spin-echo), which allow multiplicity editing of 13C spectra, just as DEPT experiments do. However, such experiments are also frequently referred to as APT. This has been changed accordingly in the SI.

Reviewer 2 Report

This article by Kerdraon et al reports an investigation into the use of diselenides as pre-catalysts for thiol-thiol ester exchange reactions.  Overall, I found this manuscript easy to read and the authors have made a solid case for the use of selenols as catalysts.

I recommend publication in Molecules once the authors have addressed my comments below.

1)  Throughout the manuscript the authors keep referring to 7a, 17 and 18 as catalysts, they are pre-catalysts.  For example, line 117: "potencies of catalysts 17 and 7a ..." this really should be "potencies of catalysts 13 and 8a, which are formed in situ from 17 and 7a, …".

2)  Scheme 1 legend should be reworked as compounds 13 and 14 are not shown in the Scheme.

3)  The authors are incorrectly quoting the results from reference 49, see line 154, where they state: "promote the nucleophilic attack of the hydroxide ion on the selenoester …", that is the kinetics in reference 49 show a slope of 1 below the pKa of the dimethylamino group (protonation of the amino). Thus, if hydroxide was the nucleophile the slope should be +2 and therefore the nucleophile in these hydrolysis reactions is water.  The authors should correct this error.

4)  Figure 4a is hard to decipher due to the number of overlapping data points, perhaps this could be made into two separate figures (with one perhaps being moved to SI).

5)  In the SI section, the authors should give the equation that they used for the first-order fits to thioester formation, the graphs of which are labeled as Table S2!  For example, it is clear that with 100 mM of pre-catalyst 7a the fit goes through >100% conversion.

6)  There is no mention of MPAA in the SI experimental protocols but there is a graph with it as a catalyst (Table S2).

suggested changes include:

1) line 82; change "we evaluated the interest of selenol compounds 13 and 14 for catalyzing …" to "we evaluated the potential of selenol compounds 13 and 14 to catalyze …".

2) line 143; change "Another stricking observation that has been done during …" to "Another striking observation made during …"

Author Response

Dear Referee,

We thank you for helping us improving our manuscript. We highly appreciate your comments and suggestions and our response can be found below within your report in the attached .doc file.

Best Regards

Oleg Melnyk & Vangelis Agouridas
